

# A dual model node based optimization algorithm for simultaneous escape routing in PCBs

Asad Ali[1], Anjum Naveed[2] and Muhammad Zeeshan[2]

[1] National Chiao Tung University, Taiwan, Taiwan
[2] National University of Sciences and Technology (NUST), Islamabad, Pakistan

## ABSTRACT

Simultaneous Escape Routing (SER) is the escaping of circuit pins simultaneously from inside two or more pin arrays. This is comparatively difficult as compared to routing in a single array and has not been addressed by previous studies. The increase in pin array complexity has made the manual SER in PCBs a very inefficient and tedious task and there surely is need for the automated routing algorithms. In this work, we propose network flow based optimal algorithm that uses integer linear program to solve SER problem and area routing problem in two stages. In the first stage, pins are escaped to the boundaries of pin arrays simultaneously. These escaped pins are connected with each other in the second stage. The proposed algorithm is tested for different benchmark sizes of grids and the results show that it is not only better in terms of routability but also outperforms existing state of the art algorithms in terms of time consumption. The existing algorithms either fails to achieve higher routability or have larger time complexities, whereas the proposed algorithm achieves 99.9% routability and is also independent of grid topology and component pin arrangement, which shows the superiority of proposed algorithm over the existing algorithms.

## INTRODUCTION

Printed circuit boards (PCB) support and connect the electrical components and provide conduction between them. Design of PCB is one of the most important things in electrical circuit and plays a vital role in performance of the circuit. There are multiple integrated circuits (*ICs*) which are placed on the PCB and ever evolving manufacturing techniques have increased the complexity of these ICs and PCBs. The enhanced manufacturing techniques have not only reduced the size of the package but, have also exponentially increased the number of pins in an IC. The footprints of ICs have shrunk in size while the number of pins have increased. Around 2,000 pins are present in the modern day high end packages *Yan, Ma & Wong (2012)* with very small routing resources. This makes the footprint of an IC a very dense entity and manual routing becomes a hectic task in such a dense environment. Some other constraints, in addition to the PCB density,

Corresponding author
Asad Ali, ali.eed06g@nctu.edu.tw

such as planar and pairwise routing, and length matching (*Ritchey & John Zasio, 2006*; *Mitzner, 2009*) are also inflicted upon PCB routing.

These constraints combined together with the increasing density of the packages impose a bigger challenge on the PCB routing and, therefore, take manual routing out of the consideration. This is where the automated routers come into the play for the solution of these issues and are currently used for the PCB routing. Automated routers help solve various PCB routing problems including escape routing and the area routing problem. The escaping of pins to the array boundary is known as escape routing and connecting these escaped pins in the intermediate area is known as area routing. Escape routing and area routing together play very important role in PCB routing. The pins on boundary of a pin array can easily be escaped and connected to their corresponding pins by keeping the via constraint in mind. But there are many pins which are inside the pin array and they cannot be connected directly and have to be escaped towards the boundary first which is known as escape routing problem. Escape routing problem is one of the critical challenges in the PCB design because the modern ICs have a pin array that contains a large number of pins. The studies show that escape routing has three different types (*Yan & Wong, 2010*; *Yan, Ma & Wong, 2012*), namely Unordered Escape, Ordered Escape and Simultaneous Escape Routing (*SER*). Ordered and unordered escape routing consider a single pin array while two ICs are considered simultaneously in SER.

Simultaneous escape routing (SER) is relatively less explored as compared to the ordered and unordered escape routing. SER is also considerably difficult to achieve because of the reason that instead of a single IC, there are two ICs in SER and their pins need to be escaped simultaneously so that they could be connected to each other. There are very few studies on SER, however *Ozdal, Wong & Honsinger (2007)* can be considered as the pioneer as this work consider two pin arrays simultaneously for the first time and minimize the net ordering mismatch. This work generated different patterns for a single pin and then selects one of the patterns such that the mismatching in net ordering along the boundaries is minimized. They used polynomial time algorithm for smaller problems and use randomized algorithm approach for routing in the case of larger problems without the inclusion of performance constraints (*Ozdal, Wong & Honsinger, 2007*). Some heuristic solutions (*Lee, Chen & Chin, 2013*, *Chin & Chen, 2013*) have also been proposed for SER problem via using boundary and planar routing. Either too much time is consumed by these solutions or the achieved routability is less than acceptable percentage. In our recent study (*Ali, Zeeshan & Naveed, 2017*), we propose the solution to routability problem through a network flow approach where we introduce two different models; one is based on the links and second on nodes. They are introduced to solve SER problem in PCBs through optimization technique. The models are successful in routing over small grids but consumed too much time when run over the larger grids.

In this work, we extend our previous study (*Ali, Zeeshan & Naveed, 2017*) in order to solve both the problems including SER and area routing while considering time and routability constraints for smaller as well as larger grids. In our previous work, we did not consider larger grids. This research propose an optimal routing algorithms for both SER and area routing for smaller and larger grids. The proposed algorithm uses the

network flow approach to develop an integer linear program for solution of these two problems. The problems of simultaneous escape and area routing are solved in multiple stages. Two different integer linear programs are developed for these two stages. The first program simultaneously escapes the pin from inside of both the pin arrays in order to reach the boundary of pin arrays. The second program fetches the results from first program and the purpose of second program is to connect the escaped points with each other.

Existing SER solutions suffers from various challenges including fixed pattern generation and higher time complexity. Some solutions in literature also lead to the pin blockage problem and resource wastage. These studies employed randomized approaches and heuristic algorithms but fail to provide efficient solution. In this study, we provide an optimal solution having a time complexity of under a minute along with 99.9% routability for the problem of SER in the printed circuit boards. This shows that the proposed solution is better than the solutions existing in the literature in terms of routability, time complexity and computational costs. Followings are the contributions of this work:

- Mapping of the PCB routing problem to network flow routing problem;
- Proposal of an algorithm for solution of SER that uses an integer linear program and provides optimal solution;
- Proposal of an algorithm for area routing that uses an integer linear program and provides optimal solution;
- Linkage between the proposed algorithms to obtain the end-to-end routing;
- 99.9% routability for pins;
- Reduction of time complexity.

Rest of the paper is organized as: Related literature is described in the next section followed by problem formulation. Then proposed network flow approach and dual node based routing are detailed. We finally discuss results before concluding the work.

## Related work

PCBs are widely used for the modern age electric circuits fabrication. The design of a PCB plays a vital role in the performance of electric circuits. The ever evolving technology has not only reduced the size of ICs that are to be placed on the PCBs but has also increased the pin count of ICs. This makes the process of manual PCB routing a very hectic task and demands automated PCB routing. The problems that are to be solved in the PCB routing are escape routing, area routing, length matching, and number of layers that are to be used. Many studies in the literature have addressed these routing problems related to the PCBs.

Some studies have proposed the heuristics algorithms to obtain a length matching routing (*Zhang et al., 2015b*) and others have used differential pair escape (*Li et al., 2012*, *2019*), single signal escape routing (*Wu & Wong, 2013*) or both (*Wang et al., 2014*) for escape routing along with addressing the length matching problem. The most notable

technique used for the PCB routing is optimization. There are various studies in literature that have mapped different PCB routing problems to the optimization problem such as longest common interval sequence problem (*Kong et al., 2007*), multi-layer assignment problem (*Yan, Kong & Wong, 2009*), network flow problem (*Sattar & Ignjatovic, 2016*), pin assignment and escape routing problem (*Lei & Mak, 2015*), and maximum disjoint set of boundary rectangles problem for bus routing (*Ahmadinejad & Zarrabi-Zadeh, 2016*). A recent study has proposed a routing method that is based upon maze and it uses a hierarchical scheme for bus routing and the proposed method uses a rip-up and re-route technique as well in order to improve the efficiency (*Chen et al., 2019*).

There are different problems to be solved in the PCB routing as discussed earlier and one of the most important problem among these is the escape routing where the pins from inside of an IC are to be escaped to the boundary of IC. Many studies in the literature have proposed solutions for escape routing problem among which some have considered single layer (*Lei & Mak, 2015*) in the PCB while, multiple layers (*Bayless, Hoos & Hu, 2016*, *Zhang et al., 2015a*) have also been considered. There are studies that only consider escape routing (*McDaniel, Grissom & Brisk, 2014*) while, there are some studies which consider other problems like length matching (*Zhang et al., 2015b*, *Yan, 2015*, *Chang, Wen & Chang, 2019*) along with the escape routing as well. Some have used staggered pin array (*Ho, Lee & Chang, 2013*; *Ho et al., 2013*) while others have used grid pin array (*Jiao & Dong, 2016*) as well. Apart from electrical circuits, escape routing is also used for designing micro-fluid biochips PCBs (*McDaniel et al., 2016*). The most widely used technique for the solution of escape routing is the optimization theory and it has been used for the past many years and also been employed by the current studies (*Katagiri et al., 2016*; *Serrano et al., 2016*; *Ahmadinejad & Zarrabi-Zadeh, 2016*).

The problem of escape routing is not the only problem to which the optimization theory provides solution to. The optimization theory has also been deployed by several other fields in which routing of packets in the wireless networks is the most prominent one. There is a recent study in which optimization theory is used in the wireless sensor networks (WSNs) for the smart power grids (*Kurt et al., 2017*). The authors have proposed a novel mixed integer program with the objective function of maximizing the lifetime of a WSN through joint optimization of data packet size and the transmission power level. Apart from wireless sensor networks, the rechargeable sensor networks also face the problem of energy harvesting. Optimization techniques have also been used in these types of networks in order to jointly optimize the data sensing and transmission and achieve a balanced energy allocation scheme (*Zhang, He & Chen, 2015*). The optimization theory is also used in addressing the power optimization issues of the communication systems to fulfill the different demands of different message types regarding the QoS, length of packet and transmission rates ?. The optimization theory has also been used to increase the efficiency of the urban rail timetable (*Xue et al., 2019*). The authors propose a nonlinear integer program in order to obtain a rail timetable which is efficient. The model is then simplified into an integer program with a single objective. A 9.5% reduction in wasted capacity is achieved through the use of a genetic algorithm in this study.

The application of optimization theory is not only limited to the fields of communication and networking but it is also been used in the vehicle routing problem (*Thongkham & Kaewman, 2019*), network reconfiguration for Distribution Systems (*Guo et al., 2020*), microgrid systems (*Wu et al., 2019*), distributed energy resource allocation in virtual power plants (*Ko & Joo, 2020*), control of humanoid robots *Tedrake (2017)*, smart grid ecosystem (*Koutsopoulos, Papaioannou & Hatzi, 2016*) and Computation of the Centroidal Voronoi Tessellations (CVT) that is widely used in the computer graphics (*Liu et al., 2016*). *Ozdal & Wong (2004)* and *Ozdal & Wong (2006)* can be regarded as the initial work on SER as they consider two pin arrays simultaneously and minimize the net ordering mismatch. There are some studies in the literature that have proposed a simultaneous pin assignment *Xiang, Tang & Wong (2001)*, *Wang, Lai & Liu (1991)*. *Ozdal & Wong (2004, 2006)* propose a methodology to escape the pins to boundaries in such a way that crossings are minimized in the intermediate area. Polynomial time algorithm is used for smaller problems and randomized algorithm approach is proposed for routing in the case of larger examples.

It is stated that the routing resources available inside the pin array are less as compared to the routing resources that are available in the intermediate area. This is because of the reason that pins are densely packed inside the pin array. Also, using vias is not allowed within the pin array and the routing inside pin array must be free from conflicts and overlapping. Via usage is allowed in the intermediate area as opposed to the pin arrays. The problem is formulated as to find out best escape routing solution within the pin array so that the conflicts in intermediate area are minimized. There are two phases to solve the problem. In the first phase, a single layer is taken at a time and maximum possible non conflicting routes are packed on that layer. The conflicted routes are routed on the next layer and hence, escape routing is completed in this manner (*Ozdal & Wong, 2004, 2006*).

In the second phase of problem solution, a congestion based net-by-net approach is used for routing in the intermediate area. Routing conflicts are allowed at the beginning and optimal solution is found by rip-up and re-route approach. The ripping up of routes that are formed in the first phase is discouraged and in order to find a conflict free route for all pins which is obtained eventually. A number of routing patterns are defined for each net and a polynomial time algorithm is proposed in order to choose the best possible combination from all the given possible routing patterns. The results for different industrial data sets are obtained which show that optimal routes are increased and time required for routing is decreased as compared to other algorithms. Although it is a good approach, but 99.9% routability is not achieved in the escape routing problem and too much time is taken in the area routing problem (*Ozdal & Wong, 2004, 2006*).

Same authors propose an algorithm in *Ozdal, Wong & Honsinger (2007)* to solve the escape routing problems in multiple components simultaneously. Some design constraints have also been considered along with routing. The escape patterns considered in this algorithm are more generalized in comparison to the algorithm proposed in their previous research (*Ozdal & Wong, 2004*). The objective is to minimize the crossover in intermediate area, hence less via usage. For smaller circuits, results given by their

proposed randomized algorithm are within 3% of optimal solution, but optimal solution is not achieved in the case of larger circuits due to time complexity. They have also assumed fixed pattern generation which can be successful in the case of smaller circuits where components are well aligned, but complex problems cannot be solved easily by using these approaches (*Ozdal & Wong, 2004*; *Ozdal, Wong & Honsinger, 2007*). B-Escape (*Luo et al., 2010*; *Luo et al., 2011*) solves this issue by removing the consumed routing area by a pin and shrinking the boundary. B-Escape reduced the time complexity and also increased the routability but, it can cause pin blockage when certain area is removed and boundaries are shrunk. Also, the time taken to solve most of the problems is still greater than 1 min which is not optimal and needs to be reduced further than that (*Luo et al., 2010*; *Luo et al., 2011*).

In addition to the above mentioned studies, other techniques have also been used in the literature for SER solution and one of them is negotiated congestion based routing scheme (*Qiang, Yan & Wong, 2010*). Negotiated congestion based routing already has its application in FPGA routing. However, it is not being used for solving SER problem. A negotiated congestion based router is proposed which applies the negotiated congestion routing scheme on an underlying routing graph. In negotiated congestion approach, all pins are forced to negotiate for a resource which is the routing space in case of PCB. The pin needing most resources is determined. Some resources are shared between pins and these pins are routed again and again iteratively till no resources are shared among the pins. Results have been compared with Cadence PCB router Allegro and it was found that proposed router is able to achieve 99.9% routability in case of such circuits which are not routed completely by the Allegro. Time consumption is also less, but the issue is that there are some examples in which the proposed router is not able to achieve 99.9% routability while Allegro is able to achieve. It is proposed that the router should be used in collaboration with Allegro so that all the problems can be solved.

There is another research (*Yan, Sung & Chen, 2011*) which uses ordered escape routing in order to solve the SER problem. The SER problem is solved by determining the net order. A bipartite graph is created on the basis of location of escape pins and transformation of net numbers is done. After doing SER, the net numbers are recovered. Basically, the orders of escape nets are found along the boundary in such a way that there is no crossover in the intermediate area. This approach achieves 99.9% routability along with reduction in time consumption, but the achieved net order is not routable in all cases. This approach can fail in some specific cases where the ordering proposed by the approach is not routable. Quite recently, net ordering has been incorporated with the SER through proposal of a novel net ordering algorithm in order to reduce the running time (*Sattar et al., 2017*). Recently, we have proposed the solution to this problem through a network flow approach (*Ali, Zeeshan & Naveed, 2017*). We proposed the solution to this problem through a network flow approach where we introduced two different models; one is based on the links and called Link based routing, while the other is based on nodes and is called Nodes based routing. They are introduced to solve SER problem in PCBs

through optimization techniques. The models were successful in routing over small grids but consumed too much time when run over the larger grids.

As per our findings, the SER has been explored by these studies and they have provided some good solutions with various limitations. There are some major issues that still seem to be unresolved and they include fixed pattern generation and time complexity. Most of these studies are unable to solve problems like pin blockage and resource under utilization. The studies have used randomized approaches and heuristic algorithms and optimal solutions are not provided. In this study, we provide an optimal solution having a time complexity of under a minute along with 99.9% routability for the problem of SER in the PCBs.

## Problem formulation

The number of pins in an IC has been increased considerably and the footprint of an IC has shrunk in recent years making the escape routing problem very difficult. The studies in literature have used randomized approaches and heuristic algorithms to solve the escape routing problem which have not been able to reduce time complexity, achieve 99.9% routability, and provide optimal solutions. The problem of SER has been solved in this study through proposal of an optimal algorithm that not only achieves the 99.9% routability but also consumes very less time and memory to find out the optimal solution.

The problem of SER basically boils down to connection of pins of two components. There are many components on a PCB but there are also many layers available for routing of these components. We solve this problem by taking two components at a time and producing the best possible routing solution for them. The routing of two components can be generalized for multiple components. The basic problem is to escape the pins from inside the pin array of two components and then to connect them by using area routing. As the size of a pin array is very small and routing resources are very less inside the pin array so vias cannot be used inside the pin array and all the pins inside pin array must be routed on a same layer. Once these pins are escaped onto the boundary, different layers can be used for their routing to avoid the conflicts in area routing.

The problem is formulated such that these conflicts in the intermediate area are minimized and least possible vias are used. The problem can be understood with the help of illustrations. Three ICs can be seen in the Fig. 1, in which some pins are represented with lines inside a hollow circle. Different patterns of the lines inside hollow circles can be seen. The pins represented by same patterned circles have to be connected with each other. The pins represented by the simple lines inside the hollow circle have to be connected with the pins represented by the simple lines inside the hollow circle and the pins represented by the crossed lines inside the hollow circle have to be connected with the pins represented by the crossed lines inside the hollow circle. Before connection, these points have to be escaped towards the boundary first as shown in Fig. 2.

It can be seen in the Fig. 2 that all the pins have been escaped to the boundary of pin arrays without any conflicts inside the pin array, but there is a conflict route in the intermediate area as shown in Fig. 3. In this situation, one of these routes will be routed on

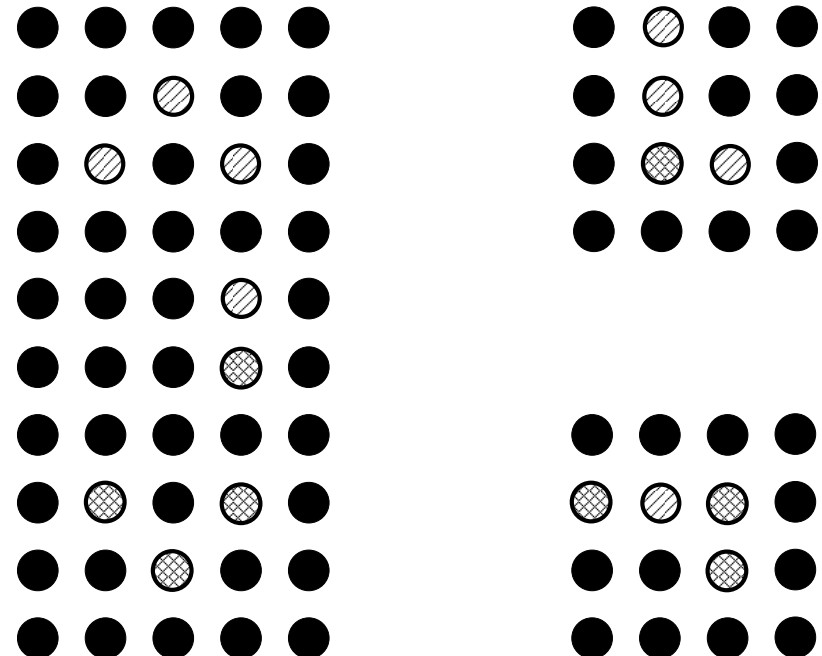

**Figure 1 ICs that need to be connected.** 

the same layer while the other will be left to be routed on the second layer as shown in Fig. 4. This shows that the basic problem is to escape the pins from inside of the pin array on the same layer and then connect these escaped pins by using minimum possible layers in the intermediate area. For this purpose, the pins will be escaped to the boundary in such a way so that conflicts are minimized in the intermediate area. For the sake of simplicity, routing is done only on a single layer. So the problem can be stated as:

> "To simultaneously escape the pins from inside the pin arrays to the boundaries and connect the escaped pins while achieving 99.9% routability and minimizing the time required".

## Network flow approach

The problem formulated in the previous section was to escape the pins from inside the pin arrays to the boundary of pin arrays and then to connect them in the intermediate area. The components whose pins are to be connected are placed on a board in a manner which is shown in the Fig. 5. The square boxes and remaining points in the Fig. 5 represent the ICs and grid points for intermediate area routing respectively. The Objective is to connect these pins with each other by first escaping them from inside of the IC simultaneously and then connecting the escaped pins together. This issue is similar to the traffic flows in a network and hence, it can be mapped to a network flow problem. In the network, a node needs to communicate with another node and hence there is a need to establish a route between them. Similarly, the pins in our problem can be considered as nodes of a network the connection between them can be considered as a route. The

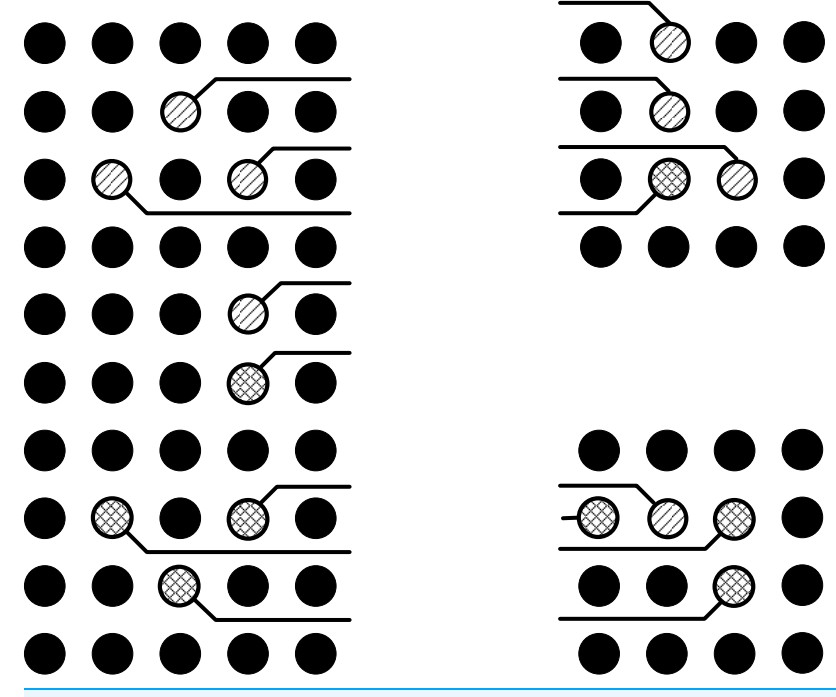

**Figure 2 Points escaped to the boundary.**

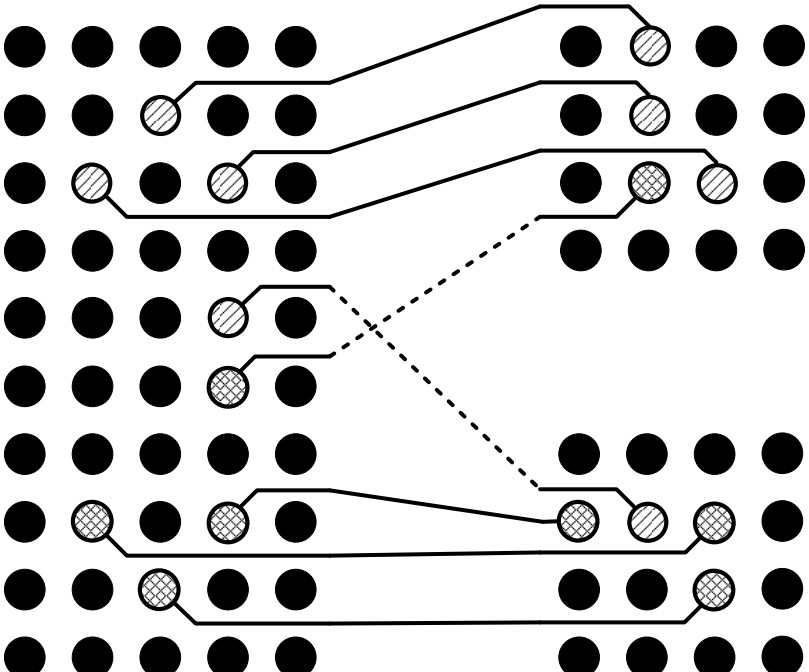

**Figure 3 Conflict route in the intermediate area.**

purpose is to create a route from one node to another node i.e. connect two pins with each other. A grid is shown in Fig. 5 and this grid is similar to a network grid and there multiple nodes in that network in a mesh topology.

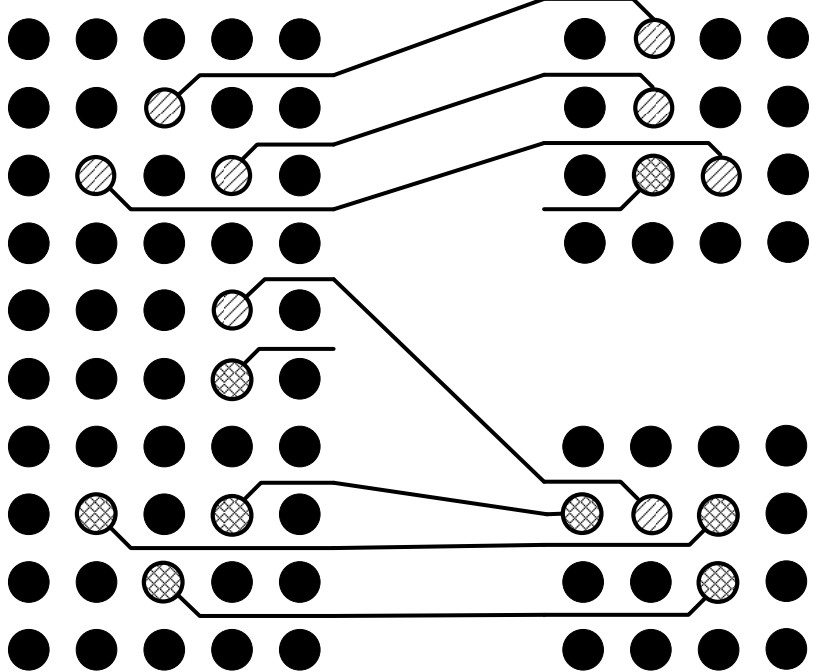

**Figure 4  Routing in the intermediate area.**

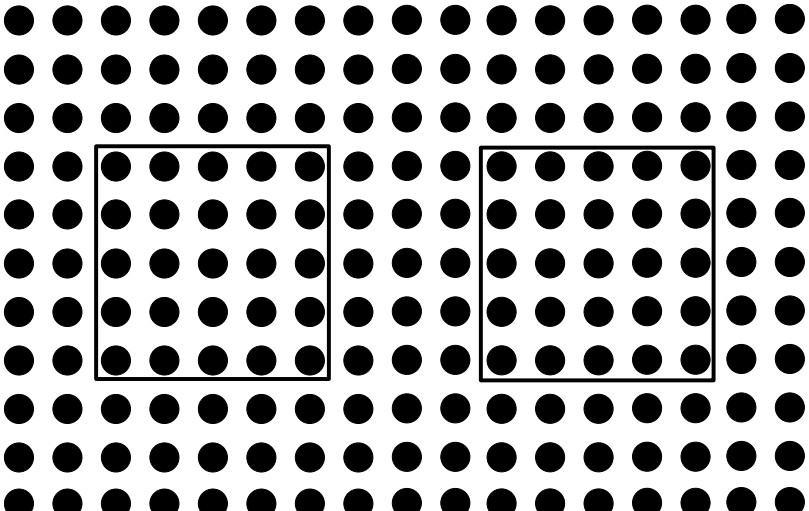

**Figure 5  ICs placed on PCB.**

Different types of pins are shown in Fig. 6 with the help of circular points where the pins that need to be connected are shown with a hollow circle and crossed lines inside the circles. The filled circles show the intermediate nodes through which these pins can be connected to each other. In terms of the network flow, these cross lined hollow circles are considered as the source and destination nodes while the filled circles are considered as

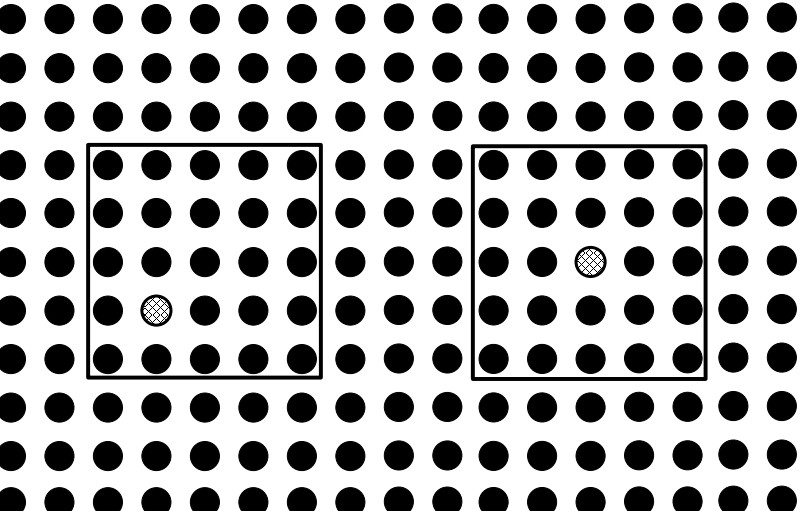

**Figure 6 Network flow approach.**

**Table 1 Mapping Table.**

| PCB routing | Network flow |
| --- | --- |
| Pins | Network Nodes |
| Pins that need to be connected | Source node and destination node |
| pin-pin pair | Network flows |
| Connection of two neighboring pins | Network Links |
| Layers | Channels |

the intermediate nodes. The objective is to create a route from the source node towards the destination node by making use of the intermediate nodes. There can be multiple pins that need to be connected with each other and hence, there can be multiple source-destination pairs and these pairs are considered as flows in the network.

These pairs of pins can be mapped to a network flow. The immediate neighbors of a source and destination pin are also mapped to the neighboring network nodes. As there is a link between network nodes, this link is considered as the pin connection. The Table 1 shows how the mapping from the routing in a PCB to flows in a network is carried out. The terms related to PCB routing are mentioned in the PCB Routing column and their mapping to network flow is shown in the Network Flow column.

## DUAL MODEL NODE BASED ROUTING

The problem of SER and area routing has been solved in two stages with the help of two different integer linear programs. We name this approach as dual model node based routing. The dual model node based routing has two stages and these stages are named as local routing and global routing. The pins of pin arrays are escaped to the boundary of pin arrays in the first stage which is known as local routing. These escaped pins are then

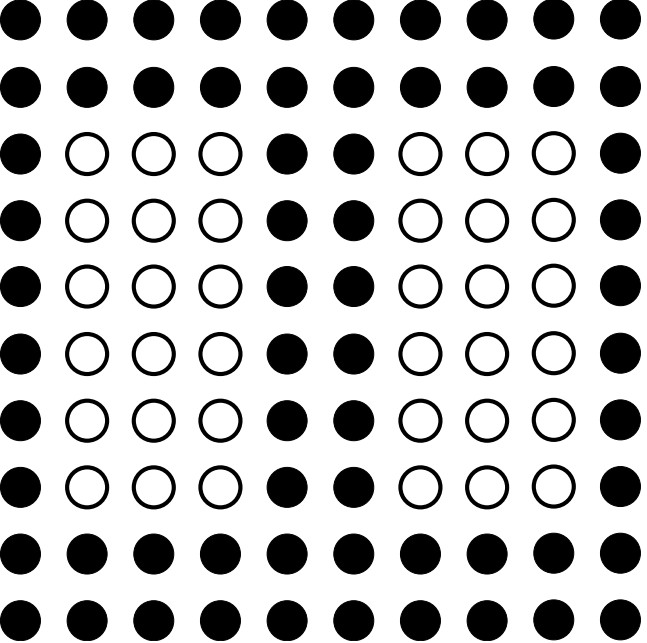

**Figure 7 The grid used in the Fourth model.**

connected with each other in the second stage which is known as global routing. The concept of local and global routing is explained as follows:

- Local Routing,
- Global Routing.

This idea of local and global routing of this approach can be understood with the help of Fig. 7:

**Local routing:** The grid was divided into two parts as it can be seen in the Fig. 7. The hollow circles represent the points that are used in the local routing. Idea is to escape the points that are to be connected to one of the boundary points. The filled circles on the boundary of hollow circles are the boundary points. When the desired points have selected a boundary point each, then these boundary points are saved and their coordinates are passed to the second model through a script. The script checks for those points for which boundary points are selected and then saves those boundary points as source and destination points for the second model. This is called as the Local routing.

**Global routing:** The work of first stage known as local routing ends after escaping the desired points to the boundary points. The second model starts working from there and the source and destination boundary points are connected through second model of this approach. This is called the Global routing. End-to-end routing with less memory consumption is ensured by these two models together.

### Addressing link based and node based routing issues

Memory consumption is the main issue with the link based and node based routing proposed in our earlier work (*Ali, Zeeshan & Naveed, 2017*). There are large number of the

---

**Algorithm 1  Local routing model.**

1: Decision variable = X [a,b]

2: Number of variables = a × b

  *So in the first model of this approach, the decision variable is:*

3: X [flows, connecting points]

  *For the 10 × 10 grid:*

4: Flows = 1, connecting points = 36

  *So the number of variables became:*

5: Variables = flows x connecting points

6: Variables = 1 × 36 = 36

---

**Algorithm 2  Global routing model.**

1: Decision variable = X [a,b]

2: Number of variables = a × b

  *So in the second model of this approach, the decision variable is:*

3: X [flows, connecting points]

  *For the 10 × 10 grid:*

4: Flows = 1,connecting points = 64

  *So the number of variables became:*

5: Variables = flows × connecting points

6: Variables = 1 × 64 = 64

---

variables used in those approaches and as the number of grid points increase, the memory consumption also increases. These issues are solved by the dual model node based routing proposed in this paper which not only divides the total memory consumption between two models known as local routing and global routing but, also ensures the solution with unique routes. For a simple 10 × 10 grid, the number of variables used in the dual model node based routing can be calculated by using Algorithm 1. Local routing model of the dual model node based routing used 36 variables. The number of variables in the global routing model can be calculated a presented in Algorithm 2.

The total number of the variables in both models is 100 which is equal to the number of variables used in the node based routing proposed in our previous study (*Ali, Zeeshan & Naveed, 2017*) but, these variables are divided among two models and are solved separately. Hence there is no issue of memory consumption and the end to end route with uniqueness is also ensured by the dual model node based routing. The mathematical model and the related terminology of dual model node based routing is explained in the next section.

**Table 2 Terminology used in the mathematical models of the dual model node based routing.**

| Term | Meaning |
| --- | --- |
| LF | The set which includes sources of local points in grid |
| N(LF) | The set which includes neighboring points of LF points |
| N(P) | The set which includes neighbors of all P points |
| SD | The set which includes source and destination points of Local flows |
| SDBP | The set which includes source and destination boundary points of Local flows |
| D(LF) | The set which includes the destination flows |
| P | The set which includes all grid points |
| BP | The set which includes all grid boundary points |
| N(BP) | The set which includes boundary points neighbors |
| CP | The set which includes connections of points in LF |
| startx | The parameter which has the x coordinate of boundary points selected as Local Flow's source |
| starty | The parameter which has the y coordinate of boundary points selected as Local Flow's source |
| endx | The parameter which has the x coordinate of boundary points selected as Local Flow's destination |
| endy | The parameter which has the y coordinate of boundary points selected as Local Flow's destination |
| X[flows, points] | The binary decision variable for selection of a flow |
| Y[flows, edges] | The binary decision variable for selection of a boundary point for a flow |

**Algorithm 3 Local routing model.**

1: The source and destination points must be selected

2: One neighbour of the source and destination point must also be selected

3: If a point is selected, other than source and destination, then two of its neighbors must also be selected

4: Not more than two neighbours of a point must be selected

5: At least one boundary point for each flow must be selected

6: At least one boundary point for the destination of each flow must be selected

7: A point should not be selected for more than one flow

8: Subject to: $\min \sum\limits_{(i,j)\varepsilon SD} \sum\limits_{(l,m)\varepsilon P} X_{i,j,l,m}$

## Terminology and mathematical model

The Table 2 shows the terminology that is helpful in understanding the dual model node based routing.

The local routing algorithm of the dual model node based routing is explained in Algorithm 3 and it uses the constraints which are detailed in Algorithm 3. A flowchart for Algorithm 3 is shown in Fig. S1.

## Constraints for local routing model

The objective function of Local Routing Model is as follows:

$$\min \sum_{(i,j)\varepsilon SD} \sum_{(l,m)\varepsilon P} X_{i,j,l,m}$$

subject to:

$$X_{i,j,i,j} = 1, \forall (i,j)\varepsilon LF \tag{1}$$

$$\sum_{(l,m)\varepsilon N(LF)} X_{i,j,l,m} = 1, \forall (i,j)\varepsilon LF \tag{2}$$

The Eq. (1) makes sure that the starting point for a particular flow is selected. The Eq. (2) makes sure that one of the neighbors of that starting point must be selected. This is used in order to start the routing from source and keep it moving towards its neighbor.

These Eqs. (1) and (2) start the flow but do not make sure that it will continue towards the boundary points. We have used three equations for that. The Eqs. (3), (4) and (5) ensure the continuity of flow towards the boundary points and also make sure that a neighbor which is already selected, does not get selected again. This is used to make sure that flow does not go back towards source.

$$2 \times X_{i,j,l,m} \leq \sum_{(a,b)\varepsilon N(l,m)} X_{i,j,a,b}, \forall (i,j)\varepsilon LF, \forall (l,m)\varepsilon CP - LF \tag{3}$$

$$\sum_{(c,d)\varepsilon N(l,m)} X_{a,b,c,d} \leq 2, \forall (a,b)\varepsilon LF, \forall (l,m)\varepsilon CP \tag{4}$$

$$\sum_{(l,m)\varepsilon BP} X_{i,j,l,m} = 1, \forall (i,j)\varepsilon LF \tag{5}$$

Now, we have the routed the source points towards the boundary points but we also need an equation that routes the same destination points to the boundary points. Therefore, we use the same equations for the destination points which were used for the source points. The Eqs. (6) and (7) make sure that the destination points of the flow are selected along with one of their neighbors. This is done to make sure that the route starts from destination point and move towards one of its neighbors.

$$X_{a,b,a,b} = 1, \forall (i,j)\varepsilon LF, \forall (a,b)\varepsilon D(LF) \tag{6}$$

$$\sum_{(l,m)\varepsilon N(a,b)} X_{a,b,l,m} = 1, \forall (i,j)\varepsilon LF, \forall (a,b)\varepsilon D(LF) \tag{7}$$

The Eqs. (8), (9) and (10) and (11) ensure the continuity of flow of destination point towards the boundary points. The previously mentioned equations only selected the destination point and one of the neighbors but, these equations make sure that an already

| Algorithm 4 | Global routing model. |

1: The source boundary point must be selected

2: The sum over all the neighboring boundary points of the source boundary point must be less than one

3: If a boundary point is selected, other than source and destination boundary points, then two of its neighboring boundary points must also be selected

4: Not more than two neighbors of a boundary point must be selected

5: The destination boundary point must also be selected

6: A boundary point should not be selected for more than one local flow

selected neighbor is not selected again and the route continues to flow towards the boundary points.

$$2 \times X_{i,j,l,m} \leq \sum_{(a,b)\varepsilon N(l,m)} X_{i,j,a,b}, \forall (l,m)\varepsilon CP - D(LF), \forall (i,j)\varepsilon D(LF) \tag{8}$$

$$\sum_{(c,d)\varepsilon N(l,m)} X_{a,b,c,d} \leq 2, \forall (a,b)\varepsilon D(LF), \forall (l,m)\varepsilon CP \tag{9}$$

$$\sum_{(l,m)\varepsilon BP} X_{a,b,l,m} = 1, \forall (a,b)\varepsilon D(LF) \tag{10}$$

$$\sum_{(i,j)\varepsilon SD} X_{i,j,a,b}, \forall (i,j)\varepsilon LF, \forall (a,b)\varepsilon CP \tag{11}$$

## Constraints for global routing model

The global routing algorithm of the dual model node based routing is explained in Algorithm 4 and it uses the following objective function and constraints in Eqs. (12) & (13). A flowchart for Algorithm 4 is shown in Figs. S2.

objective function:

$$\max \sum_{(i,j)\varepsilon LF} \sum_{(c,d)\varepsilon N(startx[i,j],starty[i,j])} Y_{i,j,c,d}$$

The local routing model has selected the boundary points for both the source and destination points. The local routing model has provided these boundary points to the global routing model. Now, it is the responsibility of the global routing model to connect these boundary points together. The first step is to select these boundary points and the Eqs. (12) and (13) ensure that the boundary points provided by the local routing model are selected in the global routing model.

subject to:

$$Y_{i,j,startx[i,j],starty[i,j]} = 1, \forall (i,j)\varepsilon LF \tag{12}$$

$$Y_{a,b,endx[a,b],endy[a,b]} = 1, \forall (a,b)\varepsilon LF \tag{13}$$

After the selection of boundary points, the next step is to select one of the neighbors of the source boundary points and then continue the flow towards the destination boundary points. The Eqs. (14), (15) and (16) are used to ensure the continuity of the flow from source boundary point to destination boundary point.

$$\sum_{(c,d)\varepsilon N(startx[i,j],starty[i,j])} Y_{a,b,c,d} \leq 1, \forall (a,b)\varepsilon LF \tag{14}$$

$$2 \times Y_{a,b,l,m} \leq \sum_{(c,d)\varepsilon N(l,m)} Y_{a,b,c,d},$$

$$\forall (a,b)\varepsilon LF, \forall (l,m)\varepsilon BP - SDBP(LF) \tag{15}$$

$$\sum_{(c,d)\varepsilon N(l,m)} Y_{a,b,c,d} \leq 2, \forall (a,b)\varepsilon LF, \forall (l,m)\varepsilon BP \tag{16}$$

There is a possibility that two or more than two flows will select a same point in their route. This possibility will cause the crossover and uniqueness of the routes will be affected. These chances of flows crossover in the second model is avoided by Eq. (17).

$$\sum_{(i,j)\varepsilon LF} Y_{i,j,a,b} \leq 1, \forall (a,b)\varepsilon BP \tag{17}$$

The issues of memory consumption and the route uniqueness are solved by the dual model node based routing. Apart from that, very small time was consumed to find out the optimal path. In the next section, some results are included to support the models proposed.

## RESULTS AND DISCUSSIONS

The dual model node based routing is composed of two separate stages. The pins are simultaneously escaped to the boundary of the pin arrays in the local routing stage and these escaped pins are connected with each other in the global routing stage. A Mathematical Programming Language (AMPL) is used to write the script of the algorithm and Gurobi solver is used to solve the algorithm. There are also other solvers available apart from Gurobi but Gurobi is the best among all of them in terms of calculation time and optimality. The dual model node based routing is based upon Integer Linear Programming approach and for Integer Linear Programming approach, Gurobi, Bonmin, and Minos are a few good choices. Gurobi has been preferred over the others because of time complexity and integrity issues.

The integrality is relaxed by the Minos solver in order to produce optimal results and this loss of integrality is not recovered at the output due to which the results are not desirable in some cases. Integers are replaced with decimal numbers in some cases which is not feasible for the proposed model as integrality must be maintained. Strict integrality is maintained by the Bonmin solver, on the other hand, which is also not feasible for the proposed model as too much time is taken to find out the optimal solution. The solution of these two problems is provided by Gurobi. The integrality is relaxed by Gurobi during the

problem solution and it is recovered when the results are generated. In this way, less time is consumed to find optimal results with strict integrality. The time consumption of each solver depends upon the complexity of the problem. We selected Gurobi through hit and trial because different solvers are suitable for different problems. In our case, Gurobi is the best possible selection as it retains the integrity and also takes less time as compared to other solvers.

These characteristics of Gurobi solver make it a perfect choice for solution of dual model node based routing. AMPL is chosen for execution of the algorithms because freedom of high level implementation is provided by it. A wide range of optimization problems can be solved with the help of AMPL and same syntax is used for declaration of data, model and commands. The local routing and global routing models are written in a model file which reads the data from a separate data file. Two models are included in the proposed algorithm and these two models are connected together with the help of a script that gathers values from the output of the local routing model and provides these values to the input of the global routing model.

The proposed algorithm was written in AMPL and solved with the help of Gurobi solver. The Gurobi solver and AMPL software were run on two different machines in order to have a comparison of algorithm running time on different machines. The first machine used for algorithm execution was an Intel Core 2 Duo PC with a processor of 2.10 GHz and a RAM of 2 GB. The other machine used for the execution of the algorithm was NEOS server which can be accessed on-line. NEOS server consisted of a 2.8 GHz (12 cores) CPU-2x Intel Xeon X5660, 64 GB RAM, and 2x 500 GB/2 TB SATA drives disk. The purpose of using these two machines was to show that the proposed algorithm can be used by PC users as well as server operators. We have mentioned specifications of both the machines so that anyone can easily use the same specifications and check the validity of results and also compare the results with their proposed solution. There was definitely a difference between time consumed by these two machines which is mentioned in the next section.

The dual model node based routing is tested on three girds of different sizes. These three grids were of small, medium and large sizes respectively. The intention was to run the proposed model on grids of different sizes so that efficiency of model could be claimed for all types of grids. The specifications of three grids against which the model was tested are as follows:

- Small Grid,
- Medium Grid,
- Large Grid.

## SMALL GRID

The first grid has a size of 20 × 20 and according to the dual model node based routing, this grid is divided into two parts. These two parts are solved with the local routing model first and then with the global routing model. The first part contains two 13 × 6 grids which

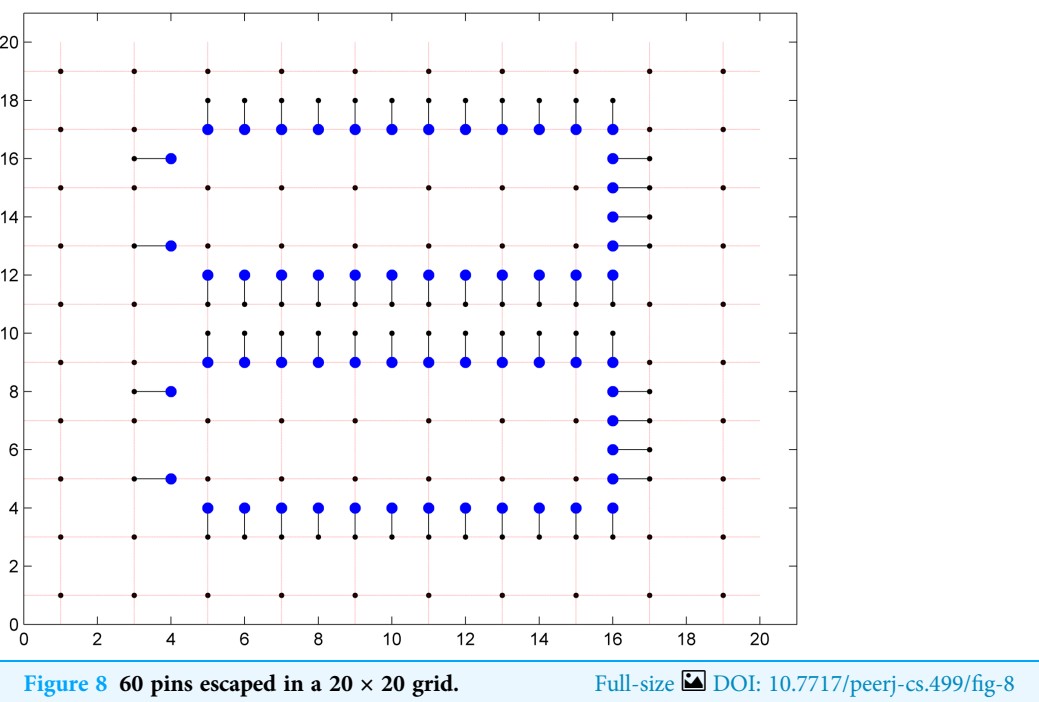

**Figure 8  60 pins escaped in a 20 × 20 grid.**     

are used for escaping the points to the boundary and this is solved by the local routing model. The second part contains the remaining points which are used for the area routing and it is solved by the global routing model. A total of 60 pins have been escaped through this grid in 0.4 s when solved on a Intel core 2 duo PC and 0.32 s when solved through NEOS server on-line. The escaped points in the 20 × 20 grid are shown in the Fig. 8. These escaped pins are connected through the second model as shown in Fig. 9.

It can be seen that not all the escaped pins are connected by the global routing model. It is because of the reason that not enough routing resources are available to route all these escaped pins on a single layer. There are two solutions to this, either the remaining escaped pins can be routed on other layers or the grid can be expanded sideways in order to route all the escaped pins on a single layer. The second solution is not feasible because the solver used for solving this problem is an academic version of Gurobi which cannot handle more than a specified number of constraints. The commercial version of Gurobi needs to be purchased in order to solve the larger grid for the area routing problem. Therefore, it has been shown with the help of a relatively smaller grid that area routing can be done with the help of the proposed model. The proposed dual model node based routing is equally good for area routing of the large grids.

## MEDIUM GRID

The results of the dual model node based routing have also been compared with a study in literature *Wu & Wong (2013)*. The solver used by *Wu & Wong (2013)* is min-cost flow solver CS2. Although, the solver used by them is different from our solver but the choice of a solver depends upon the model and in our case, Gurobi proved to be the best solver in all cases. In *Wu & Wong (2013)*, network flow model has been proposed for escaping the

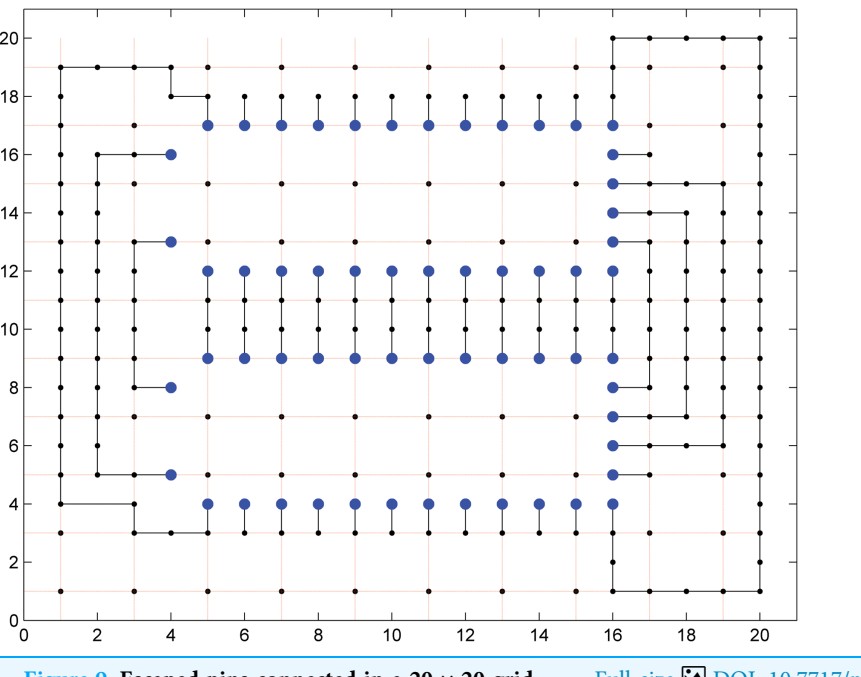

**Figure 9 Escaped pins connected in a 20 × 20 grid.**

pins to the boundaries and few results have been obtained in this study using industrial data. In one of their cases, 42 pins were escaped in a grid of 11 × 27 in 0.08 s. We designed a 30 × 30 grid in which the first model was a 11 × 27 grid similar to the study *Wu & Wong (2013)*. We were able to escape 68 pins in the same grid within 0.12 s through NEOS server. This shows that all the desired pins have been escaped to the boundary. The memory used for the processing was 68.48 MB. The memory consumption is very less and the time taken to solve the problem of routing is also quite less. The results of escaped pins are shown in Fig. 10. After this, the global routing model will connect these escaped pins but it has not been shown because of the reason mentioned in the previous sub section that the academic version of the gurobi cannot handle more than a specified number of constraints. If we use another solver instead of Gurobi, we would not be able to get optimal results. Also, we do not have enough resources to buy the commercial version of Gurobi. We have already tested the global routing model for the smaller cases in order to show that our model works perfectly well.

## LARGE GRID

The large grid taken to evaluate the dual model node based routing is a 50 × 50 grid. The results have been compared with another case of the same study *Wu & Wong (2013)*. In this case, 86 and 112 pins were escaped in a grid of 17 × 34 in 0.16 s. We designed a 50 × 50 grid in which the local routing model contained 17 × 34 grid similar to this case in the study *Wu & Wong (2013)*. We were able to escape 112 pins in the same grid within 0.02 s through NEOS server. This shows that the all the desired pins are escaped to the boundary within a minimal time. The memory consumed during processing was 190.6 MB which is also quite less. The number of escaped pins can also be increased by

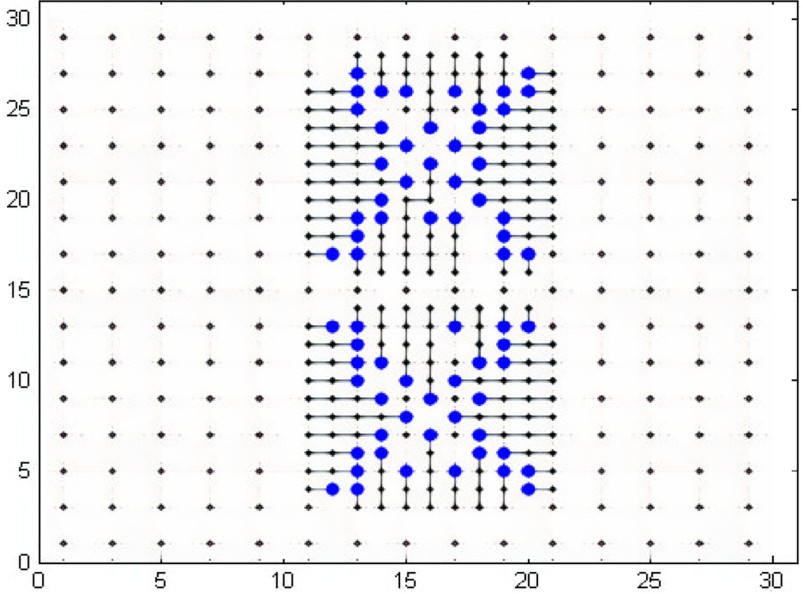

**Figure 10 Sixty-eight pins escaped in a 30 × 30 grid.**

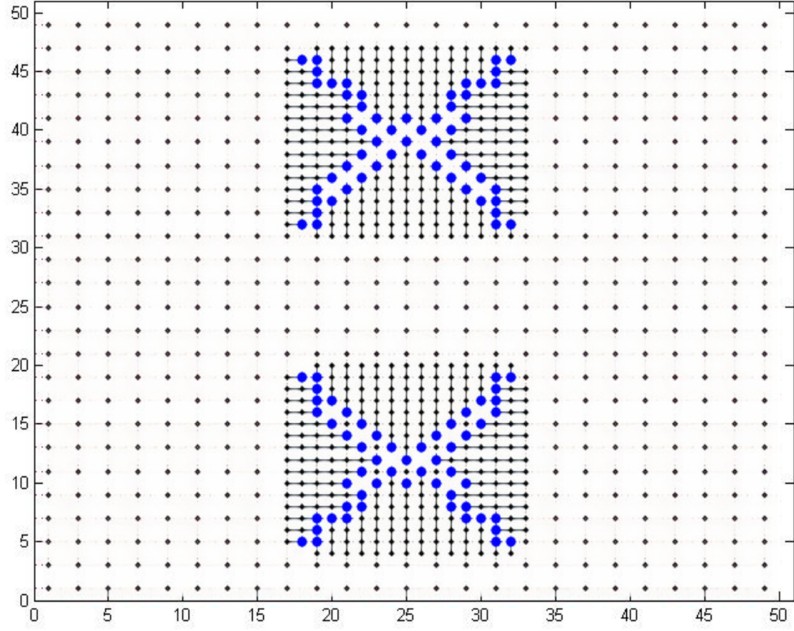

**Figure 11 One hundred and 12 pins escaped in a 50 × 50 grid.**

changing our grid topology a bit which is out of scope of this work. The results of escaped pins are shown in Fig. 11.

The case used for comparison uses only a 17 × 34 grid and we use a 50 × 50 grid in which we escape 112 points using an internal grid of 17 × 34. If we use only a grid of 17 × 34 and run first model only, then we can get results in half time i.e., 0.01 s and memory consumption is also reduced to 97.05 MB. This is a considerable reduction in time as

**Table 3 Time and memory consumption for different grids.**

| Grid | Array | Pins escaped | Pins escaped (*Wu & Wong, 2013*) | Time | Time (*Wu & Wong, 2013*) | Memory |
|------|-------|--------------|--------------------------------|------|------------------------|--------|
| Small | $20 \times 20$ | 60 | N/A | 0.32 | N/A | Negligible |
| Medium | $30 \times 30$ | 68 | 42 | 0.12 | 0.08 | 68.48 MB |
| Large | $50 \times 50$ | 112 | 86 | 0.02 | 0.16 | 190.6 MB |

compared to the time (0.16 s) taken by the case taken in the compared study. The results of area routing in both the above mentioned cases have not been obtained due to the limitations of the academic version of the Gurobi solver. The results for small, medium, and large grid are shown in Table 3. It can be seen that as the number of pins increases, our proposed model takes less and less time as compared to *Wu & Wong (2013)*. This also shows the scalability of our model. As the number of pins is increasing, our model is taking less time and hence, it would still be suitable if number of pins is scaled up. We could not test all the cases provided in *Wu & Wong (2013)* because of academic solver's limitations and relevancy. We choose the most relevant cases from *Wu & Wong (2013)* for comparison.

Table 3 shows that the proposed dual model node based routing method is great in terms of routability and it also takes very less time and memory to provide the routing solution. We have compared our model's efficacy with state-of-the-art models and our model outperforms state-of-the-art in the terms of time consumed. The memory consumed by the proposed dual model node based routing method is also considerably small which shows that no high end machines are required to run this model.

## CONCLUSION

This work address two problem at the same that includes simultaneous escape routing and area routing in PCB. The main contributions of this research are the mapping of PCB routing problem to network flow problem and the proposal of two algorithms for SER and area routing using integer linear programs. This work also links the local and global routing algorithms in order to achieve the end-to-end routing on a PCB. The proposed algorithms are efficient in terms of routing and time consumption as they outperform the existing algorithms and achieve 99.9% routability. Currently, we are working to propose a blend of ordered escape routing with the SER for further optimizing routability and improve time consumption. We also aspire to scale this algorithm for dual layer and multi-layer PCBs.

### Funding

The authors received no funding for this work.

### Competing Interests

The authors declare that they have no competing interests.

## Author Contributions

- Asad Ali conceived and designed the experiments, performed the experiments, performed the computation work, prepared figures and/or tables, authored or reviewed drafts of the paper, and approved the final draft.
- Anjum Naveed conceived and designed the experiments, analyzed the data, authored or reviewed drafts of the paper, supervised the work, and approved the final draft.
- Muhammad Zeeshan analyzed the data, prepared figures and/or tables, authored or reviewed drafts of the paper, and approved the final draft.

## Data Availability

The code file for the model, the script to transfer data between models, and three data files are available as Supplemental Files.

## Supplemental Information

Supplemental information for this article can be found online at http://dx.doi.org/10.7717/peerj-cs.499#supplemental-information.

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
