# Peer review of "A dual model node based optimization algorithm for simultaneous escape routing in PCBs"

_PeerJ Computer Science, doi:10.7717/peerj-cs.499_

## Round 0.1 · original submission · Major Revisions

Dear Dr. Zeeshan,

Thank you for your submission to PeerJ Computer Science.
It is my opinion as the Academic Editor for your article - A dual model node based optimization algorithm for simultaneous escape routing in PCBs - that it requires a number of Major Revisions.

My suggested changes and reviewer comments are shown below and on your article 'Overview' screen.

Please address these changes and resubmit. Although not a hard deadline please try to submit your revision within the next 30 days.

With kind regards,
Xiaolong Li
Academic Editor, PeerJ Computer Science

Reviewer 1 ·

Basic reporting

no comments

Experimental design

no comments

Validity of the findings

no comments

Additional comments

The manuscript has technical contributions. Moreover, it is improved significantly but I still have some concerns.
1. There are many grammatical, typographical, and sentence construction errors here and there throughout the paper. In the following, I have listed some such problems. The authors are required to fix the rest of the similar problems and fix those before the resubmission.
• Please use one convention throughout the documents for abbreviation, i.e., Printed circuit boards (PCBs) or printed circuit boards (PCBs).
• Abbreviate one time and then use the abbreviation in the rest of the documents.
• Use one convention for referencing or citing Algorithms 1 or algorithm1
• There are many complex sentences in the updated version of the manuscript, please make them small and clear to understand the meaning, i.e., First sentence of the last paragraph of page 8.
• Use the same pattern in the paper, i.e., use comma before every “and” if using at the start.


2. Please compare your model efficacy with state-of-the-art model, i.e., when simultaneous Escape Routing is escaping of circuit pins simultaneously from inside two or more pin arrays
3. Please provide or figure in which you compare Gurabi, Minos and Bonmin in terms of time and integrity, which will strengthen your selection.
4. Please provide figures in the results section that show time consumption and memory consumption.
5. Authors claim that they achieve 100% routability but statistically it is not the correct statement. You can write 99.9% but not 100%.
6. Please plot some graphs to check the scalability of your proposed techniques.
7. I suggest to please add flowchart also so it would be easy to understand your work.


Concluding Remarks: This paper may be accepted after incorporating minor comments by the author

Reviewer 2 ·

Basic reporting

1. INTRODUCTION states clearly what are the context, challenges and contributions
2. Quite sufficient relevant works and their limitations are posed and well-organized for the problem formulation of this paper.
3. Figures such as Figure 10 and 11 are suggested to use better quality in its resolution.
4. In RESULTS AND DISCUSSIONS, result statistics of test cases are better to be made into a table for a clear view of comparison with others’ previous works.
5. In INTRODUCTION and RELATED WORK, cross usage of abbreviations and full names are not preferred, such as PCBs vs. Printed Circuit Boards and SER vs. Simultaneous Escape Routing.

Experimental design

1. In DUAL MODEL NODE BASED ROUTING, algorithms are well explained and unambiguous.
2. The algorithm was designed in a way to ensure 100% routability with its proper constraints.
3. The lack of evaluation of the global routing model in both Medium Grid and Large Grid cases make the conclusion weak in Line 443-445 and Line 461-462.
4. In Line 386-391 in RESULTS AND DISCUSSION, lack of provision of statistics from different solvers Bonmin and Minos as a proof of the advantages of utilizing Gurobi solver.

Validity of the findings

1. Not all cases provided in Wu and Wong(2013) were tested in this paper, it is hard to conclude that the proposed algorithm outperforms over the one in Wu and Wong(2013) as stated in Line 463-464.
2. Different solver used than Wu and Wong(2013), the comparison of experiment results don’t have exactly the same base line.
3. The paper claims significant improvement in both the memory consumption and runtime in comparison to the previous work Wu and Wong(2013), however, the machine specification used on NEOS server was not specified.
4. The contribution stated in Line 90-91 is not convincing as over half of the cases are not tested for global routing model in Line 436-466.

Additional comments

The paper sounds promising in the beginning in INTRODUCTION and the background study is quite sufficient. The overall algorithm design is proper and unambiguous. However, there is a room for improvement in the experimental design. The experimental design makes the contribution of the paper unconvincing due to the incompletion of case testing on global routing and the test environment difference between the one in others’ work.

If academic version of Gurobi solver has a limitation in constraints, you may need to consider other solvers for a full comparison in both the local and global routing models instead of leaving the global routing model untested for over half of the test cases. Or it would be more convenient if you could borrow the commercial license from some other institutes. The lack of test results makes the conclusion and contribution really weak.
And the machine specification is also a concern when comparing the runtime with others.

·

Basic reporting

The research work is interesting and is organized well.
The manuscript require professional language editing.

Experimental design

The design and analysis is sound

Validity of the findings

Abstract and conclusion need to be rewritten for easy and clear understanding.

Additional comments

The research work is interesting and is organized well.
The manuscript require professional language editing.
Abstract and conclusion need to be rewritten for easy and clear understanding.

---

## Round 0.2 · accepted · Accept

Dear Dr. Zeeshan,

Thank you for your submission to PeerJ Computer Science.
I am writing to inform you that your manuscript - A dual model node based optimization algorithm for simultaneous escape routing in PCBs - has been Accepted for publication. Congratulations!

This is your official letter of acceptance.

Congratulations again, and thank you for your submission.

With kind regards,
Xiaolong Li
Academic Editor, PeerJ Computer Science